# Adaptation of *Brucella melitensis* Antimicrobial Susceptibility Testing to the ISO 20776 Standard and Validation of the Method

**DOI:** 10.3390/microorganisms10071470

**Published:** 2022-07-20

**Authors:** Alina Tscherne, Enrico Mantel, Talar Boskani, Sylwia Budniak, Mandy Elschner, Antonio Fasanella, Siri L. Feruglio, Domenico Galante, Christian G. Giske, Roland Grunow, Judit Henczko, Christin Hinz, Wojciech Iwaniak, Daniela Jacob, Agnieszka Kedrak-Jablonska, Veronica K. Jensen, Tone B. Johansen, Gunnar Kahlmeter, Viviana Manzulli, Erika Matuschek, Falk Melzer, Maria S. Nuncio, Joseph Papaparaskevas, Ana Pelerito, Margrete Solheim, Susanne Thomann, Athanasios Tsakris, Tara Wahab, Marcin Weiner, Lothar Zoeller, Sabine Zange

**Affiliations:** 1Bundeswehr Institute of Microbiology, Neuherbergstrasse 11, 80937 Munich, Germany; alina.tscherne@viro.vetmed.uni-muenchen.de (A.T.); enricomantel@bundeswehr.org (E.M.); christinhinz@bundeswehr.org (C.H.); lotharzoeller@gmail.com (L.Z.); 2Division of Virology, Department of Veterinary Sciences, LMU Munich, Veterinärstrasse 13, 80539 Munich, Germany; 3Public Health Agency of Sweden, Nobels väg 18, 171 82 Solna, Sweden; talar.boskani@cepheid.com (T.B.); tara.wahab@folkhalsomyndigheten.se (T.W.); 4National Veterinary Research Institute, 57 Partyzantów Avenue, 24-100 Pulawy, Poland; sylwia.budniak@piwet.pulawy.pl (S.B.); akedrak@piwet.pulawy.pl (A.K.-J.); mpweiner@piwet.pulawy.pl (M.W.); 5Friedrich-Loeffler-Institut, Federal Research Institute for Animal Health, Institute of Bacterial Infections and Zoonoses, Naumburger Strasse 96a, 07743 Jena, Germany; mandy.elschner@fli.de (M.E.); falk.melzer@fli.de (F.M.); 6Istituto Zooprofilattico Sperimentale della Puglia e della Basilicata, 20 Manfredonia Street, 72121 Foggia, Italy; antonio.fasanella@izspb.it (A.F.); domenico.galante@izspb.it (D.G.); viviana.manzulli@izspb.it (V.M.); 7Norwegian Institute of Public Health, Lovisenberggata 8, 0456 Oslo, Norway; siri.laura.feruglio@fhi.no (S.L.F.); veronicaklausmark.jensen@fhi.no (V.K.J.); tone.johansen@fhi.no (T.B.J.); margrete.solheim@fhi.no (M.S.); 8Division of Clinical Microbiology, Department of Laboratory Medicine, Karolinska Institutet, Alfred Nobels Allé 8, Huddinge, 141 52 Stockholm, Sweden; christian.giske@ki.se; 9Highly Pathogenic Microorganisms, Centre for Biological Threats and Special Pathogens, Division 2 (ZBS 2), Robert Koch Institute, Seestrasse 10, 13353 Berlin, Germany; grunowro@rki.de (R.G.); jacobd@rki.de (D.J.); 10National Public Health Center, Albert Flórián út 2-6, 1097 Budapest, Hungary; henczkoj@gmail.com; 11CEDROB S.A., Ujazdowek 2A, 06-400 Ciechanow, Poland; iwaniak.wojciech@wp.pl; 12EUCAST Development Laboratory (EDL) for Bacteria, 351 85 Växjö, Sweden; gunnar.kahlmeter@kronoberg.se (G.K.); erika.matuschek@kronoberg.se (E.M.); 13National Institute of Health, Av. Padre Cruz, 1649-016 Lisbon, Portugal; sofia.nuncio@insa.min-saude.pt (M.S.N.); ana.pelerito@insa.min-saude.pt (A.P.); 14Microbiology Department, Medical School, National and Kapodistrian University of Athens, Mikras Asias 75, 11527 Athens, Greece; ipapapar@med.uoa.gr (J.P.); atsakris@med.uoa.gr (A.T.); 15Laboratory of Spiez, Austrasse, 3700 Spiez, Switzerland; susanne.thomann@gmx.net

**Keywords:** *Brucella melitensis*, antimicrobial susceptibility testing, interlaboratory validation, ISO 20776 standard, CLSI

## Abstract

Brucellosis, mainly caused by *Brucella* (B.) *melitensis*, is associated with a risk of chronification and relapses. Antimicrobial susceptibility testing (AST) standards for *B. melitensis* are not available, and the agent is not yet listed in the EUCAST breakpoint tables. CLSI recommendations for *B. melitensis* exist, but they do not fulfill the requirements of the ISO 20776 standard regarding the culture medium and the incubation conditions. Under the third EU Health Programme, laboratories specializing in the diagnostics of highly pathogenic bacteria in their respective countries formed a working group within a Joint Action aiming to develop a suitable method for the AST of *B. melitensis*. Under the supervision of EUCAST representatives, this working group adapted the CLSI M45 document to the ISO 20776 standard after testing and validation. These adaptations included the comparison of various culture media, culture conditions and AST methods. A Standard Operation Procedure was derived and an interlaboratory validation was performed in order to evaluate the method. The results showed pros and cons for both of the two methods but also indicate that it is not necessary to abandon Mueller–Hinton without additives for the AST of *B. melitensis*.

## 1. Introduction

Human brucellosis caused by *Brucella* species belongs to the most common bacterial zoonotic diseases worldwide, with around 500,000 cases annually [1], being endemic in the Mediterranean basin, the Middle East, parts of Central and South America, Africa and Asia. *B. melitensis* is the predominant species causing most of the human cases [2]. In the EU, brucellosis case numbers have remained stable since the beginning of the EU-level surveillance in 2007, with the highest rate being in 2008 (735 cases) and the lowest rate being in 2019 (310 cases) [3]. Greece, Portugal, Italy, Spain, France and Germany reported the highest numbers, with between 65 and 211 human cases per year [4,5]. In these countries, brucellosis cases are either imported or caused by the consumption of unpasteurized dairy products leading to local outbreaks [6,7,8,9].

Brucellosis treatment requires long-term antibiotic therapy to prevent relapses and chronification [10,11]. Due to the intracellular and slow-growing nature of *B. melitensis*, a combination therapy including at least one antimicrobial substance with good cellular penetration is required in order to avoid treatment failures. A combination of doxycycline and aminoglycosides (gentamicin or streptomycin) or the combination of doxycycline and rifampicin supplemented by gentamicin in complicated cases is recommended [12,13,14,15]. Alternative regimens include trimethoprim–sulfamethoxazole and fluoroquinolones (mainly ciprofloxacin) [12,14,16,17]. In patients suffering from neurobrucellosis, the addition of ceftriaxone is suggested [18,19].

Still, *B. melitensis* is a frequently reported cause of bacterial laboratory infections, and cultivation requires biosafety level 3 laboratory conditions [20,21]. Thus, many clinical laboratories refuse to perform the antimicrobial susceptibility testing (AST) of *B. melitensis*. Until now, antibiotic resistance has been rare, and treatment failures have mostly been associated with non-compliance during long-term oral treatment or due to insufficient tissue penetration by antimicrobials that are active in vitro. However, mutations associated with antimicrobial resistance have been reported, e.g., *rpoB* mutations leading to phenotypic resistance towards rifampicin [22,23,24], which underlines the need for routine AST to ensure the proper selection of antibiotics for treatment. Because *B. melitensis* is considered to be a category B bioterrorism agent, engineered antimicrobial resistance is a not-too-far-fetched concern, which makes testing capacity part of the preparedness efforts.

The assessment of wild type (WT) antimicrobial susceptibility patterns and the identification of resistant phenotypes is hampered by the lack of a generally accepted AST standard and breakpoints for the antibiotics used against brucellosis, which in practice has led to a heterogeneity in AST methods and breakpoints. It is, therefore, difficult to interpret and compare the AST results between countries or even laboratories. *B. melitensis* is a fastidious organism; in particular, for the initial isolation from clinical specimens, blood-containing culture media and 5% CO_2_ incubation are strongly recommended [25]. Therefore, most studies apply the gradient strip method using Mueller–Hinton agar with 5% sheep blood for AST [26,27]. Some groups have determined MICs by broth microdilution (BMD), with supplemented Mueller–Hinton broth [28,29] or agar dilution (AD) using *Brucella* agar [30]. An AST guideline for the BMD of *Brucella* spp. is available from the CLSI (M45) using *Brucella* broth (BB), a rich culture medium adapted to its fastidious nature [31]. Nevertheless, various problems have been identified in this guideline: (i) the unavailability of breakpoints for rifampicin or fluoroquinolones; (ii) for other antimicrobials, only the category “susceptible” is defined, but no breakpoints are provided to define resistance; (iii) the trimethoprim–sulfamethoxazole MIC values cluster around the breakpoint, frequently leading to “intermediate” or even “resistant” results among WT strains; (iv) the rifampicin MIC values are unexpectedly high, although the whole-genome sequencing of several respective isolates revealed no genotypic alteration in the loci associated with rifampicin-resistant phenotypes [24,32,33].

Under the third EU Health Programme, a working group aiming at the development of a standard operation procedure (SOP) for *B. melitensis* AST was included in the European Joint Action EMERGE (efficient response to highly dangerous and emerging pathogens at the EU level), which lasted from June 2015 to January 2019. The network’s 40 laboratories specialize in highly pathogenic agents and/or represent the respective national reference laboratories for brucellosis. One goal was to adapt the CLSI method [31] to the ISO 20776 standard [34] with the support of representatives from the EUCAST Development Laboratory. EUCAST recommends that the choice of medium for AST (liquid and solid) is based on the investigation of the need for moving from Mueller–Hinton without additives to Mueller–Hinton with additives, and if this is still not sufficient for good growth, to use Fastidious Anaerobe Agar (or broth). The ISO 20776 standard is the general basis for the EUCAST recommendations, but as EUCAST develops methodology for so-called “fastidious organisms”, media are always tried in the order listed above. In order to address this topic regarding *B. melitensis*, the working group evaluated a range of culture media and methods which were applicable for *Brucella*. Eight partner laboratories validated the identified culture medium in comparison to the CLSI method in an interlaboratory validation. Finally, modifications of the incubation conditions were tested, and the final SOP was validated with *B. melitensis* WT-isolates.

## 2. Materials and Methods

### 2.1. Bacterial Strains and Culture Conditions

*B. melitensis* strain Bm150048 was isolated in 2015 from a blood culture of a patient suffering from osteomyelitis. The species identification was performed using IS7111 and *Brucella* Bruce-ladder PCR [35]. All of the experiments were conducted using Bm150048 and *B. melitensis* reference strain ATCC 23456 in parallel. A total of 57 *B. melitensis* clinical isolates were included from the Microbiology Department of the National and Kapodistrian University of Athens, Medical School, Athens, Greece, and from the National Consultant Laboratory for *Brucella* at the Bundeswehr Institute of Microbiology, Munich, Germany. In the primary AST of these 57 isolates, there were no resistances detected towards anti-brucellosis antibiotics in the respective lab; therefore, they were designated as 57 WT-isolates in the following text. Two *B. melitensis* isolates with known resistance towards rifampicin and trimethoprim–sulfamethoxazole, respectively, were included from the National Public Health Center, Budapest, Hungary, and they were designated as two *B. melitensis* non-WT-isolates in the following text. The reference strains *Escherichia coli* ATCC 25922, *Streptococcus pneumoniae* ATCC 49619 and *Staphylococcus aureus* ATCC 43300 served as quality controls. The strains were stored at −80 °C and cultivated on Columbia blood agar (Becton Dickinson, Franklin Lakes, NJ, USA) containing 5% sheep blood for 48 h at 36 ± 1 °C with 5% CO_2_. Before use, each strain was sub-cultured once. Work involving live *B. melitensis* was performed in a biosafety level (BSL-) 3 laboratory within a class II safety cabinet.

### 2.2. Culture Media

The liquid culture media used for BMD and growth kinetics were the following: cation-adjusted Mueller–Hinton broth (CAMHB, Becton Dickinson), *Brucella* broth (BB, Becton Dickinson), H-medium (MERLIN), CAMHB containing 5% horse blood and 20 mg/L β-NAD (CAMHB-F, according to EUCAST SOP) [36], and CAMHB supplemented with 10 mL/L IsovitaleX (CAMHB-X, Thermo Fisher, Waltham, MA, USA). The in-house media were autoclaved for 15 min at 121 °C. Supplements were added when the media had cooled down; afterwards, the pH was adjusted to 7.0 ± 0.2. The solid culture media for agar dilution were prepared by adding 15 g/L agar to the culture media mentioned above.

### 2.3. Correlation of the McFarland Standard and Bacterial Cell Counts

*E. coli* ATCC 25922 and *S. aureus* ATCC 43300 were suspended in 0.9% NaCl to McFarland from 0.4 to 0.6. The corresponding OD_600_ values were determined, and tenfold 0.9% NaCl dilutions were performed. Of these dilutions, 100 µL was streaked on Columbia blood agar plates and incubated for 24 to 48 h at 36 ± 1 °C with 5% CO_2_. Subsequently, the colonies were counted and the number of CFU/mL in the undiluted suspension was calculated. As the McFarland of 0.4 to 0.6 matched perfectly with the expected OD_600_ values (from 0.10 to 0.16) using *E. coli* and *S. aureus*, for *B. melitensis*, only the McFarlands of 0.4 to 0.6 without OD_600_ values were determined to reduce the risk of contamination within the BSL-3 facility.

### 2.4. Broth Microdilution Method

BMD tests were performed with user-defined commercial microdilution plates (MICRONAUT, MERLIN Diagnostika, Berlin, Germany) including the following antimicrobials and concentrations: gentamicin (from 0.004 to 8 mg/L), streptomycin (from 0.008 to 16 mg/L), ciprofloxacin (from 0.002 to 4 mg/L), levofloxacin (from 0.002 to 4 mg/L), doxycycline (from 0.004 to 8 mg/L), rifampicin (from 0.004 to 8 mg/L), and trimethoprim–sulfamethoxazole (from 0.016/0.29 to 16/304 mg/L). For quality control, the number of CFU in the bacterial inoculum was determined with the target of 5 × 10^5^ CFU/mL in the final culture broth. Therefore, the inoculum was diluted 1:1000 in 0.9% NaCl and streaked on Colombia agar plates. A range of 20 to 200 CFU/per plate was accepted. Before using a new batch of plates and/or culture broth, validation was performed with *E. coli* ATCC 25922 (BB, CAMHB) and *S. pneumoniae* ATCC 49619 (BB) by determining the MIC endpoints after incubation in CAMHB after 24 h at 36 ± 1 °C in ambient air, and for BB after 48 h at 36 ± 1 °C with 5% CO_2_. The results were compared to the corresponding QC tables (EUCAST QC tables and CLSI M45, respectively) [31,37]. For the BMD method of *B. melitensis*, in brief, 200 µL 1:10 diluted McFarland 0.5 suspension was transferred to 11 mL of culture broth, 100 µL were added to each well and MIC endpoints were read visually using an inverted mirror after incubation for 48 h at 36 ± 1 °C with 5% CO_2_, if not stated otherwise. The culture broth used in the respective experiment is stated in the corresponding section. The following variations of incubation conditions were tested: In order to test different incubation times, the plates were read after 18 h, 24 h, 36 h, 48 h and 64 h. In order to test the influence of the CO_2_ content on BMD, the plates were prepared in duplicate and incubated in ambient air and with 5% CO_2_ in parallel. In order to test the influence of the *B. melitensis* bacterial inoculum, BMD was performed with McFarland 0.5 suspension vs. 1:10 diluted McFarland 0.5 suspension for the inoculation of the respective culture broth for AST.

### 2.5. Agar Dilution Method

The agar dilution method was carried out using agent-dependent two-fold dilution concentrations: doxycycline (from 0.004 to 8 mg/L), rifampicin (from 0.004 to 8 mg/L), trimethoprim–sulfamethoxazole (from 0.016/0.29 to 8/152 mg/L), streptomycin (from 0.008 to 16 mg/L) and gentamicin (from 0.004 to 8 mg/L) (all from Sigma-Aldrich). The antimicrobial reagents were diluted and dissolved according to the EUCAST Definitive Document [38] with BB agar and CAMHB agar. The inocula were adjusted to McFarland 0.5 in 0.9% NaCl; 1 µL was spotted onto culture plates and incubated for 48 h with 5% CO_2_, and the MIC values were determined as the lowest concentration with no visible growth.

### 2.6. Growth Curve Analysis

CAMHB, CAMHB-X, CAMHB-F and BB were inoculated in triplicate with 10 CFU/mL bacterial cells and incubated at 36 ± 1 °C with 5% CO_2_. At 0 h, 19 h, 24 h, 27 h, 33 h, 43 h, 48 h, 51 h, 67 h, 72 h, 92 h, and 164 h, an aliquot of 100 µL was taken from each culture, and serial tenfold dilutions in 0.9% NaCl solution were prepared. In total, 100 µL of each dilution was streaked on Columbia blood agar plates and incubated for 48 h at 36 ± 1 °C with 5% CO_2_. Subsequently, the colonies were counted and the numbers of CFU/mL were determined.

### 2.7. Interlaboratory Validation

*B. melitensis* Bm150048 was distributed to eight EMERGE partners in an infectious substance category A transport. Interlaboratory validation (ILV) was performed with the BMD method as described above, with BB and CAMHB in parallel. BB and CAMHB were inoculated from the same bacterial McFarland 0.5 suspension. The same batch of the BMD plates was used, and the culture broths were ordered from the same manufacturer (Becton Dickinson). A reading guide and a data entry mask were provided to standardize the reporting of the results. Each institute performed 10 replicates, and the respective MIC values for each antimicrobial were reported.

### 2.8. Data Analysis

The data were prepared using GraphPad Prism version 5 (GraphPad Software Inc., San Diego, CA, USA), and were analyzed with respect to the culture media and antimicrobial substances. For ILV, data from the different sites were merged and the modal MICs were calculated for each antimicrobial substance/culture medium combination. The percentage of modal MICs plus one two-fold dilution interval on either side of the mode was calculated for each combination. When two adjacent concentrations displayed similar frequencies the mode was assumed to be somewhere between the even log_2_ concentrations that were tested, and a four-dilution range was proposed. This range of three to four dilutions was defined as the reference range for this antimicrobial substance. An interlaboratory agreement of >95% of the MIC values within this range was expected. The number of antimicrobials per culture medium fulfilling this definition was compared to the respective one, and the antimicrobials causing variances were identified.

## 3. Results

### 3.1. CFU of the Bacterial Inoculum and Its Impact on the MIC Endpoints

Bacterial cell counts corresponding to a McFarland standard range of 0.4 to 0.6 and the corresponding OD_600_ values (from 0.10 to 0.16) were determined. For *E. coli* and *S. aureus*, cell counts of McFarland 0.5 matched the expected amount of 1.5 × 10^8^ cells per mL. For *B. melitensis* the measured cell counts were 10 times higher (Appendix A). Therefore, the BMD was performed in triplicates with *B. melitensis* reference strain ATCC 23456 using undiluted McFarland 0.5 suspension (in the following, this is referred to as “undiluted”) vs. 1:10 diluted suspension (in the following, this is referred to as “1:10 dilution”) for inoculation (Figure 1). No differences in the MIC values were observed for rifampicin, gentamicin or streptomycin. For doxycycline, the 1:10 dilution led to one log_2_ step lower MIC values, and for levofloxacin and ciprofloxacin they led to more than one log_2_ step lower MIC values. For trimethoprim–sulfamethoxazole, the 1:10 dilution led to more than two log_2_ steps lower MIC values. Consequently, a 1:10 dilution of the McFarland 0.5 suspension was included to the BMD SOP in order to avoid artificially high MIC values due to a too-high bacterial count in the inoculum.

### 3.2. Comparison of the MICs in Different Culture Media

*B. melitensis* strains ATCC 23456 and Bm150048 were cultivated in five different culture media; subsequently, BMD was performed with these media (Figure 2a,b; Appendix A). BMD using CAMHB was applicable for *B. melitensis*, as enough bacterial growth was observed after 48 h to read the plates. Both strains showed comparable results among the media for most of the agents. The culture medium with the greatest difference compared to BB was CAMHB when trimethoprim–sulfamethoxazole was tested, showing five log_2_ dilution step lower MICs, with slightly less effect when supplements like IsovitaleX or horse blood were added. The fastidious culture broth from the plate manufacturer (MERLIN), H-medium, showed even higher MIC values for trimethoprim–sulfamethoxazole, and therefore was excluded in the following.

### 3.3. B. melitensis Growth Curves in Different Culture Media

The growth kinetics in CAMHB, BB, CAMHB-F and CAMBH-X were investigated for *B. melitensis* strains ATCC 23456 and Bm150048. ATCC 23456 was able to grow in all four culture media (Figure 3). The best growth was obtained for BB. CAMHB was only slightly inferior after 48 h (6 × 10^7^ CFU/mL vs. 1.07 × 10^8^ CFU/mL), whereas the growth rates in CAMHB-F and CAMHB-X were considerably lower (1.44 × 10^7^ CFU/mL vs. 3.92 × 10^7^ CFU/mL). After 72 h of incubation, the measured CFU/mL were identical for BB, CAMHB and CAMHB-X, whereas CAMHB-F showed 10-times-lower CFU/mL. After 164 h, the cultures reached the stationary phase in all of the media with identical CFU/mL. The results were reproduced with *B. melitensis* strain Bm150048 in BB and CAMHB (Appendix A). Pure CAMHB seems to be an acceptable alternative to BB for AST. The applied supplements to CAMHB showed no benefit with regard to bacterial growth. Further validation experiments were performed, therefore, with BB and pure CAMHB in parallel.

### 3.4. Broth Microdilution vs. Agar Dilution with BB vs. CAMHB

BMD and AD using CAMHB and BB were performed in parallel with *B. melitensis* strains ATCC 23456 and Bm150048 (Figure 4, Appendix A). Rifampicin’s results were within two log_2_ dilutions for both methods. For both strains and methods, the shift of the trimethoprim–sulfamethoxazole MIC values using BB in comparison to CAMHB was visible, as observed before in this study (merged data for CAMHB: 0.016 to 0.125 vs. merged data for BB: 0.5 to 4 mg/L). In conclusion, using AD, no methodological problems could be identified in BMD leading to false high trimethoprim–sulfamethoxazole or rifampicin MIC values.

### 3.5. Impact of the Incubation Time on the MIC Values

BMD was performed with BB and CAMHB, and the MIC endpoints were read after 18 h, 24 h, 42 h, 48 h, and 64 h. The earliest time points at which the MIC endpoints could be reliably read were 24 h for BB and 42 h for CAMHB (Appendix A). From this time point onwards, the MICs increased by no more than one log_2_ dilution step. Using CAMHB, it was difficult to assess the growth already after 24 h because the bacterial pellets in the plates were small and no turbidity was visible by eye.

### 3.6. Impact of the Cultivation Atmosphere on the MIC Values

The influence of the CO_2_ content on the BMD MIC values was tested in duplicate with *B. melitensis* strains ATCC 23456, Bm150048, and six clinical *B. melitensis* WT-isolates. The bacterial growth was acceptable under both conditions. Aminoglycoside’s MIC values were higher with 5% CO_2_, whereas the MIC values for rifampicin and trimethoprim–sulfamethoxazole were higher in ambient air (Figure 5a–c). Ciprofloxacin, doxycycline and levofloxacin showed identical MIC values under both conditions (Appendix A).

### 3.7. Interlaboratory Validation

ILV including eight partner sites was conducted with *B. melitensis* strain Bm150048 using BMD in 10 replicates for BB and CAMHB in parallel. In an ideal method, >95% of all of the MIC values per antimicrobial substance would lie one twofold dilution interval around the modal MIC [39], which would in this case serve as the reference range for the corresponding antimicrobial substance. CAMHB ILV showed that four out of seven antimicrobials fulfilled this prerequisite: ciprofloxacin (0.5 mg/L ± 1 log_2_ dilution step), gentamicin (0.25 mg/L ± 1 log_2_ dilution step), levofloxacin (0.5 mg/L ± 1 log_2_ dilution step) and streptomycin (0.5 mg/L ± 1 log_2_ dilution step). For three antimicrobials, the distributions of the MIC values were broader; therefore, no reference ranges could be defined [doxycycline (92%), rifampicin (91%), trimethoprim–sulfamethoxazole (65%)] (Figure 6a). BB ILV showed that even five out of seven antimicrobials fulfilled the prerequisite for the definition of the reference ranges: doxycycline (0.25 mg/L ± 1 log_2_ dilution step), ciprofloxacin (1 mg/L ± 1 log_2_ dilution step), rifampicin (2 mg/L ± 1 log_2_ dilution step), streptomycin (2 mg/L ± 1 log_2_ dilution step) and levofloxacin (0.5 mg/L ± 1 log_2_ dilution step). For two antimicrobials, the distributions of the MIC values were too broad to allow the determination of the reference ranges [gentamicin (89%), trimethoprim–sulfamethoxazole (82%)] (Figure 6b). Altogether, the percentage of the MIC values within the defined reference ranges was higher using BB compared to CAMHB. The antimicrobial substance with the highest variations in both culture media was trimethoprim–sulfamethoxazole.

### 3.8. Re-Evaluation with Clinical Isolates

Finally, 57 clinical *B. melitensis* WT-isolates were tested by means of BMD with CAMHB and BB in parallel using the new SOP. The mode MICs were calculated per antimicrobial substance for each culture medium separately, and were compared between media (Table 1). For ciprofloxacin, doxycycline, gentamicin, levofloxacin, rifampicin and streptomycin, the mode MICs were comparable, showing variances not exceeding one log_2_ dilution step from each other (mode MICs in mg/L for BB/CAMHB: ciprofloxacin 0.5/0.5, doxycycline 0.0625/0.0625, gentamicin 0.25/0.125, levofloxacin 0.5/0.5, rifampicin 1/1, streptomycin 1/0.5). The trimethoprim–sulfamethoxazole isolates showed significantly lower mode MIC values of five log_2_ dilution steps for CAMHB (mode MICs in mg/L for BB/CAMHB: trimethoprim–sulfamethoxazole 1/≤0.016).

Furthermore, the two available *B. melitensis* non-WT-isolates with known resistance towards rifampicin and trimethoprim–sulfamethoxazole, respectively, were tested with BMD using BB and CAMHB in parallel. The rifampicin-resistant isolate showed rifampicin MIC values of >8 mg/L, and the trimethoprim–sulfamethoxazole-resistant isolate showed trimethoprim–sulfamethoxazole MIC values of >16 mg/L both for BB and CAMHB.

## 4. Discussion

Constantly high numbers of brucellosis in Europe and clinicians’ demand for the in vitro assessment of antibiotic sensitivity before commencing long-term antibiotic multidrug-therapy render AST an indispensable necessity. Therefore, thoroughly elaborated standards for the testing of *B. melitensis* and its clinical breakpoints are urgently needed. B. *melitensis* is not yet listed in the EUCAST clinical breakpoint table, as it was not clear whether the ISO standard 20776 method was applicable for *B. melitensis* at all. The CLSI guideline M45 recommends the BMD method with *Brucella* broth as a culture medium. In addition to the medium, there are other issues where the CLSI differs from ISO. Its clinical breakpoints are incomplete, as brucellosis therapy-relevant antimicrobials like rifampicin, fluoroquinolones and ceftriaxone are missing. Furthermore, if the CLSI method is applied, the rifampicin MICs for *B. melitensis* WT-isolates cluster around 1 mg/L [32,33]. Although the CLSI guideline did not set a rifampicin breakpoint for *B. melitensis*, the results conflict with the currently applied rifampicin breakpoint for *Haemophilus (H.) influenzae* and *H. parainfluenzae* (CLSI M100S) [40] (S ≤ 1 mg/L) which was already used for *B. melitensis* elsewhere [41,42]. CLSI and EUCAST have not given any species a susceptible rifampicin breakpoint above 1 mg/L. Regarding EUCAST, *Helicobacter pylori* and *H. influenzae* have breakpoints of 1 mg/L. EUCAST has not defined a PK-PD (non-species related) rifampicin breakpoint [43]. Nevertheless, rifampicin proved to be effective in vivo against *Brucella* spp., although a combination therapy is always recommended. In vitro, synergy between doxycycline and rifampicin and an increased rifampicin activity at lower pH (pH 5.0) has been shown before [29]. Therefore, high MIC values might be either due to a methodological problem, or *B. melitensis* might require a higher breakpoint than other species. In order to set a breakpoint, the WT MIC distributions are needed as a basis for epidemiological cutoff values (ECOFF) plus the evaluation of the PK-PD properties of the agent. Another issue has been identified in the use of the CLSI guideline: the *B. melitensis* WT MICs for trimethoprim–sulfamethoxazole cluster around the CLSI M45 breakpoint of 2 mg/L [data from Zange S., not published] [32], or were even reported to be resistant [44]. In BB, the MICs for trimethoprim–sulfamethoxazole might be elevated due to the thymidine concentration in the rich culture broth, which has been known since the late 1970s [45]. The effect was also observable when the CLSI M45 method was validated with QC strains. The reference range for trimethoprim–sulfamethoxazole using BB could only be defined for *S. pneumoniae*, and not for the other reference strains (*E. coli* and *S. aureus*) due to an unusual variability of the obtained MICs [46]. Consequently, the lower limit of the reference range for *S. pneumoniae* was set two dilution steps higher for BB than for CAMHB (range of 0.125 mg/L to 1 mg/L for CAMHB vs. 0.5 mg/L to 2 mg/L for BB, according to CLSI M45). Therefore, changing the culture medium to one with a lower thymidine-concentration is expected to influence *B. melitensis* trimethoprim–sulfamethoxazole MICs. Other differences between CLSI and ISO pertain to culture conditions (ambient air vs. a 5% CO_2_ supplemented atmosphere) and an incubation time of 48 h. Finally, for *B. melitensis*, a recommendation for disc diffusion is missing in the M45 document.

In this study, various culture media typically used for the AST of fastidious organisms, plus pure CAMHB, were evaluated in order to find an alternative to BB. The validation was performed with respect to bacterial growth and its influence on the MIC values, especially for the above-described two antimicrobials. In order to exclude methodological problems, the gold standard of susceptibility testing, AD, was performed in parallel. The growth curves and BMD results showed that all of the selected culture media seemed suitable for *B. melitensis*. Even pure CAMHB was applicable, and bacterial growth was acceptable and even better than in CAMHB-F, although the latter is recommended for fastidious agents in the ISO 20776 [34]. Regarding the MIC values for trimethoprim–sulfamethoxazole, significantly lower values were observed with CAMHB as compared to BB or CAMHB-F. Growth curve analysis demonstrated that the lower MIC values are not due to the deficient growth of *B. melitensis* in CAMBH but rather to the nutrient content compared to BB. The AD results confirmed the findings, as the shift of MICs for trimethoprim–sulfamethoxazole was identical to the BMD results. The CFU counts of the inoculum used for BMD proved to be another factor influencing the MICs of trimethoprim–sulfamethoxazole. According to the literature, a McFarland standard of 0.5 used for BMD corresponds to an *E. coli* cell density of 1.5 × 10^8^ CFU/mL, and is equivalent to an OD_600_ of 0.13 [47]. The CFU counts of a McFarland 0.5 suspension of *B. melitensis* yielded 10 times more bacterial cells than *E. coli*. An impact of >3 log_2_ dilution steps was shown for trimethoprim–sulfamethoxazole MICs. Although some manufacturers of BMD plates advise the use of an even higher volume of the McFarland 0.5 suspension for fastidious bacterial agents in order to improve the reading of the plates (user manual, MICRONAUT Special Plates), it appears crucial for the AST of trimethoprim–sulfamethoxazole to adjust the correct CFU concentration according to ISO 20776 in order to avoid false-high MIC values.

Additionally, an effect of the culture media nutrient content on rifampicin MICs was expected. Koch et al. [48] described the influence of bacterial growth rates on rifampicin MICs, hypothesizing that the susceptibility towards rifampicin increases as the growth rate decreases, consistent with the longer drug penetration time in poorer culture media. Unfortunately, this effect was not discernible in *B. melitensis* regarding CAMHB compared to BB. Agar dilution produced results similar to BMD. The assessment of rifampicin susceptibility must therefore be regulated by adjusting the breakpoints. The MIC values of other tested antimicrobials differed by <1 dilution step between the two media, indicating that both are applicable.

Thus, pure CAMHB was chosen as an alternative culture medium to BB, and modifications of the incubation conditions were validated. Measuring the MICs at different time points showed that reducing the incubation time to less than 48 h made it very difficult to read the BMD plates. This makes the results invalid. Incubation in ambient air versus 5% CO_2_ supplementation showed acceptable growth with both approaches. As described above, the MIC values for aminoglycosides were one dilution step higher and for tetracyclines one dilution step lower with 5% CO_2_ [49]. Overall, our findings support incubating *B. melitensis* BMD plates in ambient air, even if this practice might not be applicable to other *Brucella* species; in particular, *B. abortus* requires a 5–10% CO_2_ atmosphere for growth [50].

The interlaboratory validation of BMD with BB and CAMHB in parallel at eight European *Brucella*-reference laboratories with one *B. melitensis* clinical isolate aimed to create antimicrobial substance-specific reference ranges based on the calculated mode per antimicrobial/medium combination. Comparable results were expected between the laboratories, with ≥95% of the MIC values differing no more than one dilution step from the mode. Nevertheless, results varied among the laboratories predominantly for CAMHB, which exceeded—for some of the antimicrobials—the margins. For CAMHB 57.1% (four out of seven) and for BB 71.1% (five out of seven) of the antimicrobial substances fulfilled the definition. After extensive discussion with representatives of EUCAST, we concluded that the deviating endpoint MICs reported by some of the participants were probably due to different approaches in reading the BMD. As described for other bacteria, trimethoprim–sulfamethoxazole seems to be the most challenging antimicrobial substance with respect to reading due to its trailing endpoints with the gradual fading of growth over several dilution steps [51,52]. Therefore, the distinct reporting of 80% and 100% inhibition-of-growth for trimethoprim–sulfamethoxazole was included in the reading instructions. As this was omitted from the interlaboratory validation, some partners might have reported 80% and others 100% inhibition. Subsequently, the reading instructions were updated for future tests.

## 5. Conclusions

We concluded from our data that it may not be necessary to abandon CAMHB as a culture medium for *B. melitensis* BMD. The results of parallel testing with both methods of 57 *B. melitensis* WT-isolates and single-resistant strains supported our decision. BB and CAMHB both have pros and cons, complicating the final choice of culture medium to proceed with (Table 2). A final assessment of the entire methodology was hampered by the lack of sufficient numbers of resistant isolates to ascertain the distinction between WT and non-WT strains when CAMHB is used. Nevertheless, CAMHB is suitable for many microorganisms, and is recommended by EUCAST. Because the other tested media showed no obvious advantages, the consortium decided to implement CAMHB for the BMD of *B. melitensis*. Furthermore, BMD plates should be prepared with the correct inoculum and incubated in ambient air for 48 h. This is a consensus decision by European *Brucella* reference labs that have collaborated for several years on this topic. The MIC determination of a higher number of isolates from different sites is now necessary to define WT MIC distributions in order to set clinical breakpoints.

## Figures and Tables

**Figure 1 microorganisms-10-01470-f001:**
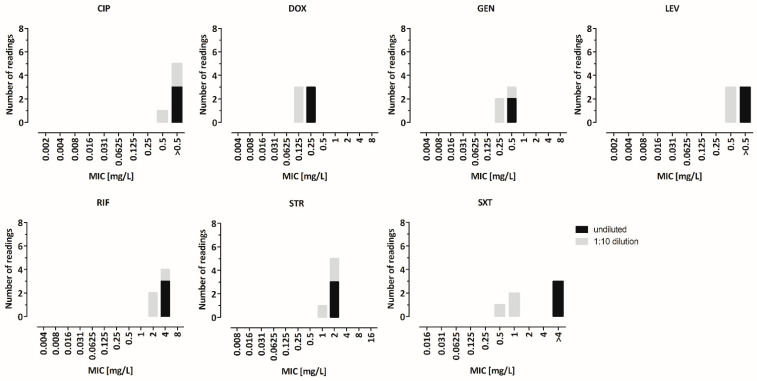
Comparison of different inoculum cell counts in the BMD. BMD was performed in triplicates with *B. melitensis* reference strain ATCC 23456 using undiluted McFarland 0.5 suspension (in the following referred to as “undiluted”) vs. 1:10 diluted suspension (in the following referred to as “1:10 dilution”) for inoculation. The MIC values were determined after incubation in a 5% CO_2_ atmosphere for 48 h. No differences in MIC values were observed for rifampicin (2 to 4 mg/L), gentamicin (0.25 to 0.5 mg/L) or streptomycin (1 to 2 mg/L). For doxycycline, the 1:10 dilution led to one log_2_ step lower MIC values (0.125 mg/L vs. 0.25 mg/L). For levofloxacin and ciprofloxacin, the 1:10 dilution led to >1 log_2_ step lower MIC values (0.5 mg/L vs. >0.5 mg/L). For trimethoprim–sulfamethoxazole, the 1:10 dilution led to >2 log_2_ dilution step lower MIC values (0.5 to 1 mg/L vs. >4 mg/L). CIP, ciprofloxacin; LEV, levofloxacin; DOX, doxycycline; GEN, gentamicin; STR, streptomycin; SXT, trimethoprim–sulfamethoxazole; RIF, rifampicin.

**Figure 2 microorganisms-10-01470-f002:**
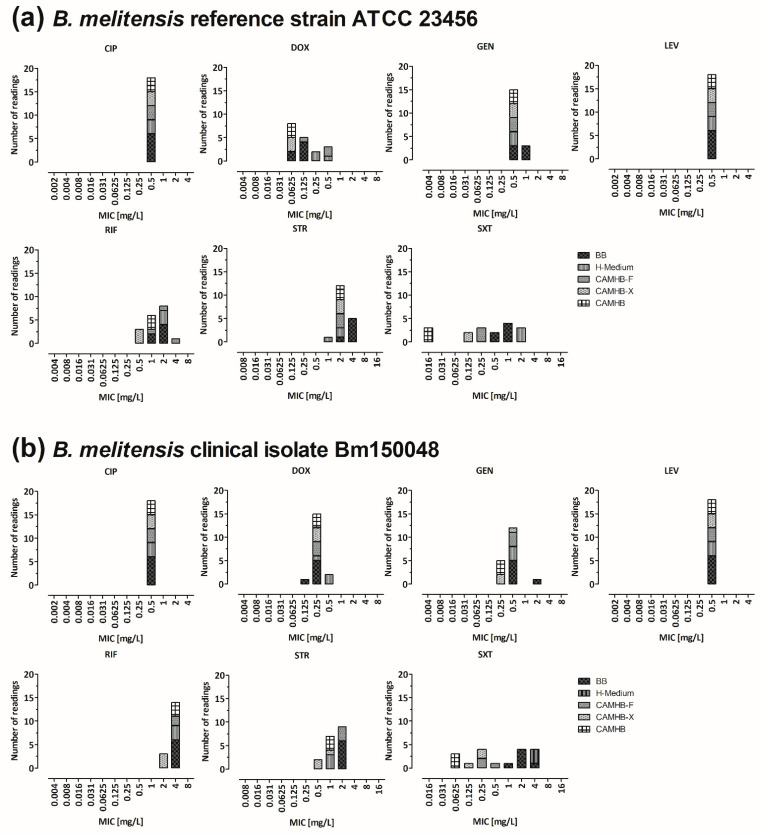
Comparison of five different culture media recommended for fastidious organisms (BB, CAMHB, CAMHB-F, CAMHB-X and H-medium). The BMD was performed in triplicates using CAMHB, CAMHB-F, CAMHB-X and H-medium, and six times for BB. The MIC values were determined after incubation in a 5% CO_2_ atmosphere for 48 h. (**a**) *B. melitensis* reference strain ATCC 23456 showed comparable MIC values for gentamicin and streptomycin for all five culture media. For trimethoprim–sulfamethoxazole, the observed MIC values showed a range of eight log_2_ dilution steps: the lowest MIC value was measured for CAMHB (0.016 mg/L), which was five log_2_ dilutions lower than for BB (0.5 and 1.0 mg/L); the other media showed one to two log_2_ dilutions lower MIC values than BB (CAMHB-X: 0.125 mg/L; CAMHB-F: 0.25 mg/L), and the highest MIC-value was observed for H-medium (2 mg/L). For doxycycline, the obtained MIC values covered a range of four log_2_ dilutions clustering around the MIC values using BB (CAMHB/CAMHB-X: 0.0625 mg/L vs. CAMHB-F: 0.5 mg/L). For rifampicin, the MIC values covered a range of four log_2_ dilutions, with similar results for BB, CAMHB, CAMHB-F and H-medium (1 to 4 mg/L) and CAMHB-X (0.5 mg/L) at the low end. (**b**) *B. melitensis* clinical isolate Bm150048 showed comparable MIC values on all five culture media for doxycycline, rifampicin and gentamicin. For trimethoprim–sulfamethoxazole, the MIC values covered a range of seven log_2_ dilutions (H-medium: 4 mg/L vs. CAMHB: 0.0625 mg/L). For streptomycin, the MIC values covered a range of three log_2_ dilutions (BB: 2 mg/L vs. CAMHB: 0.5–1 mg/L). CIP, ciprofloxacin; LEV, levofloxacin; DOX, doxycycline; GEN, gentamicin; STR, streptomycin; SXT, trimethoprim–sulfamethoxazole; RIF, rifampicin.

**Figure 3 microorganisms-10-01470-f003:**
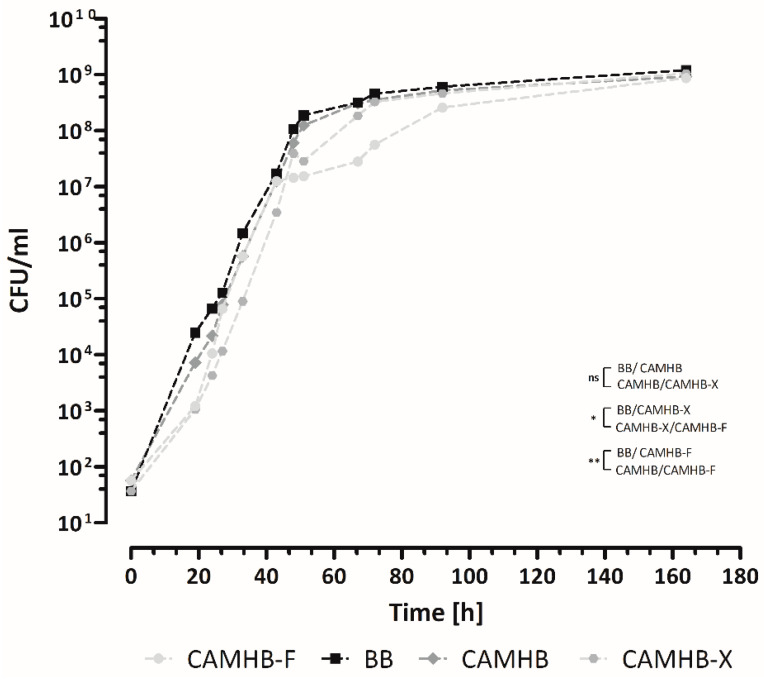
Growth curve of the *B. melitensis* reference strain ATCC 23456 in four different culture media (BB, CAMHB, CAMHB-F, CAMHB-X). Each dot represents the mean value of three replicates. The differences between the culture media were analyzed by determining the area under curve (AUC) prior to analysis by one-way ANOVA test. The error bars indicate the interquartile range (IQR) from the median. The asterisks represent statistically significant differences between groups. The best growth was obtained when using BB. CAMHB was only slightly inferior to BB after 48 h (6 × 10^7^ CFU/mL vs. 1.07 × 10^8^ CFU/mL), whereas the growth rates in CAMHB-F and CAMHB-X were considerably lower (1.44 × 10^7^ CFU/mL vs. 3.92 × 10^7^ CFU/mL). After 72 h of incubation, the measured CFU/mL were identical for BB, CAMHB and CAMHB-X, whereas CAMHB-F showed 10-times-lower CFU/mL counts. After 164 h, the cultures in all of the media had reached the stationary phase with identical CFU counts (~10^9^ CFU/mL). ns, non-significant; * *p* < 0.5; ** *p* < 0.01.

**Figure 4 microorganisms-10-01470-f004:**
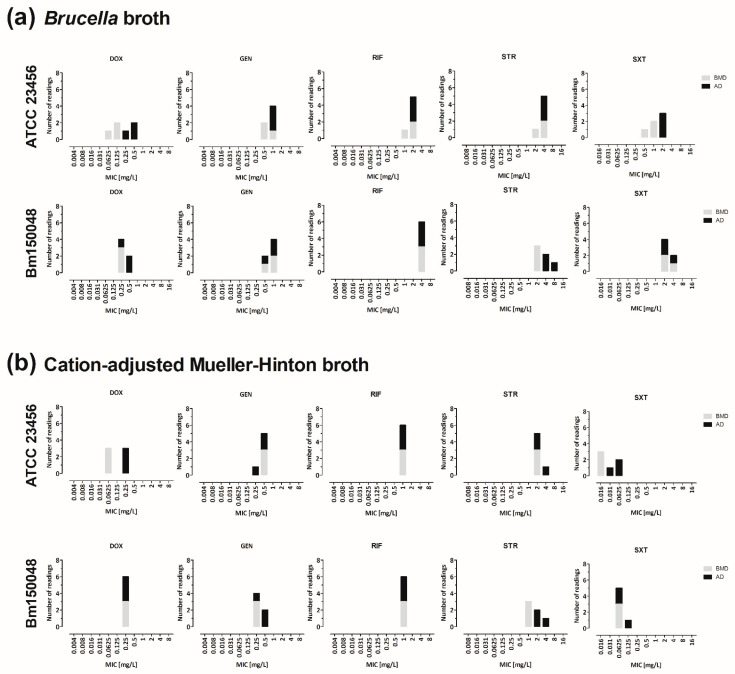
Comparison of BMD and AD using two different culture media (BB, CAMHB). *B. melitensis* reference strain ATCC 23456 and *B. melitensis* clinical isolate Bm150048 were used in triplicates to compare the MIC values obtained with BMD and AD when using BB (**a**) or CAMBH (**b**). The MIC values were obtained after incubation at 36 ± 1 °C in a 5% CO_2_ atmosphere for 48 h. The differences between the two methods were within two log_2_ dilutions for both media using ATCC 23456 towards gentamicin, rifampicin, and streptomycin. For Bm150048, the following antimicrobials fulfilled this condition: doxycycline, gentamicin, rifampicin, and trimethoprim–sulfamethoxazole. The identical antimicrobials showed variances over more than two log_2_ dilutions for the respective *B. melitensis* strain irrespective of the culture medium used: doxycycline and trimethoprim–sulfamethoxazole for ATCC 23456, and streptomycin for Bm150048. For both strains and both methods, the shift of trimethoprim–sulfamethoxazole’s MIC values using BB in comparison to CAMHB was visible, as observed before in this study.

**Figure 5 microorganisms-10-01470-f005:**
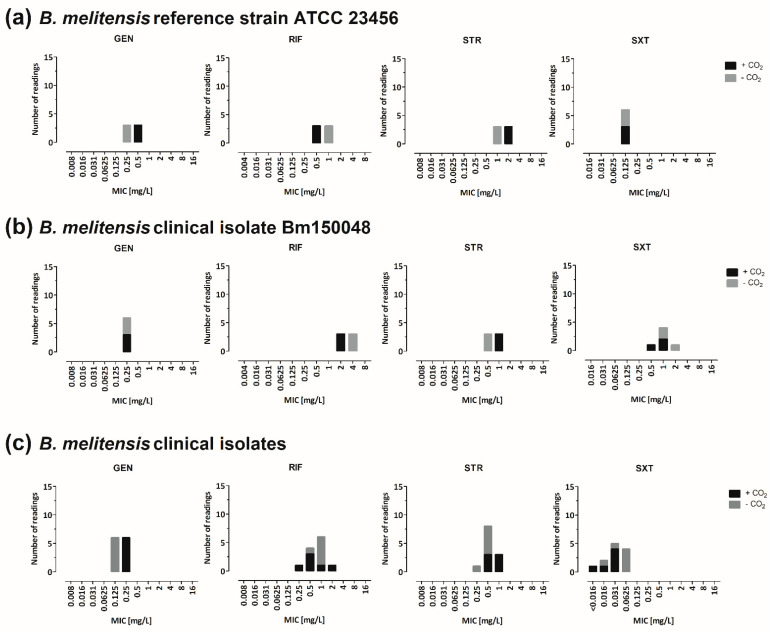
Impact of the incubation atmosphere on the MIC values. The MIC values were determined after incubation with or without 5% CO_2_ for 48 h. (**a**) *B. melitensis* reference strain ATCC 23456. (**b**) *B. melitensis* clinical isolate Bm150048. (**c**) Six clinical *B. melitensis* isolates. Aminoglycoside’s MIC values were higher with 5% CO_2_ (gentamicin: 0.125 to 0.25 mg/L vs. 0.25 to 0.5 mg/L, streptomycin: 0.5 to 1 mg/L vs. 0.5 to 2 mg/L), whereas the MIC values for rifampicin and trimethoprim–sulfamethoxazole were higher without CO_2_ (rifampicin: 0.5 to 4 mg/L vs. 0.25 to 2 mg/L, trimethoprim–sulfamethoxazole: 0.016 mg/L to 2 mg/L vs. <0.016–1 mg/L). CIP, ciprofloxacin; LEV, levofloxacin; DOX, doxycycline; GEN, gentamicin; STR, streptomycin; SXT, trimethoprim–sulfamethoxazole; RIF, rifampicin.

**Figure 6 microorganisms-10-01470-f006:**
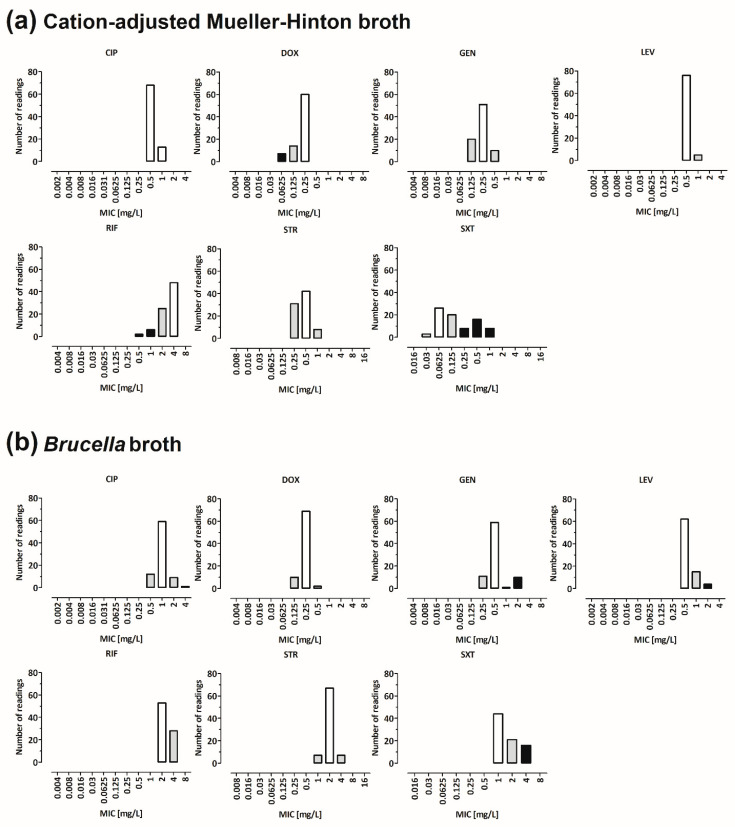
MIC values resulting from the interlaboratory validation (ILV) experiments where eight partner laboratories performed BMD with the *B. melitensis* clinical isolate Bm150048 using CAMHB (**a**). Four out of seven antimicrobials fulfilled the prerequisite to set a reference range: ciprofloxacin (0.5 mg/L ± 1 log_2_ dilution step), gentamicin (0.25 mg/L ± 1 log_2_ dilution step), levofloxacin (0.5 mg/L ± 1 log_2_ dilution step) and streptomycin (0.5 mg/L ± 1 log_2_ dilution step). For three antimicrobials, the distribution of the MIC values was broader; therefore, no reference ranges could be defined [doxycycline (92%), rifampicin (91%), trimethoprim–sulfamethoxazole (65%)]. The ILV using BB (**b**) showed that even five out of seven antimicrobials fulfilled the prerequisite for the definition of a reference range: doxycycline (0.25 mg/L ±1 log_2_ dilution step), ciprofloxacin (1 mg/L ± 1 log_2_ dilution step), rifampicin (2 mg/L ± 1 log_2_ dilution step), streptomycin (2 mg/L ± 1 log_2_ dilution) and levofloxacin (0.5 mg/L ± 1 log_2_ dilution). For two antimicrobials, the distribution of the MIC values was too broad to allow the determination of a reference range [gentamicin (89%), trimethoprim–sulfamethoxazole (82%)]. White bar, mode; grey bars, reference range; black bars, number of MIC values more than ± 1 dilution step from the mode; CIP, ciprofloxacin; LEV, levofloxacin; DOX, doxycycline; GEN, gentamicin; STR, streptomycin; SXT, trimethoprim–sulfamethoxazole; RIF, rifampicin.

**Table 1 microorganisms-10-01470-t001:** Calculated mode MICs from the BMD results of 57 clinical *B. melitensis* WT-isolates with CAMHB and BB culture medium in parallel.

Antimicrobial Substance	Medium	Mode (mg/L)
CIP	BB CAMHB	0.5 0.5
DOX	BB CAMHB	0.0625 0.0625
GEN	BB CAMHB	0.25 0.125
LEV	BB CAMHB	0.5 0.5
RIF	BB CAMHB	1.0 1.0
STR	BB CAMHB	1.0 0.5
SXT	BB CAMHB	1.0 ≤0.016

CIP, ciprofloxacin; LEV, levofloxacin; DOX, doxycycline; GEN, gentamicin; STR, streptomycin; SXT, trimethoprim–sulfamethoxazole; RIF, rifampicin.

**Table 2 microorganisms-10-01470-t002:** Summary of the pros and cons of using CAMHB instead of BB as a culture medium for the AST of *B. melitensis*.

Cation-Adjusted Mueller–Hinton II Broth (CAMHB)		*Brucella* Broth (BB)	
Pro	Con	Pro	Con
Growth after 96 h equal to BB	Method is currently not valid for *B. melitensis*	Tradition (established method)	High nutrient contents in BB lead to “false high” MIC values for trimethoprim–sulfamethoxazole
MIC values for trimethoprim–sulfamethoxazole are four log_2_ steps lower	Reading of plates is more difficult	Reading of plates easier → fewer and less discrepancies in interlaboratory validation trial	MIC values for rifampicin cluster around the applied breakpoint (S ≤ 1 mg/L)
Culture medium recommended by EUCAST			
Already used in the laboratory for other bacteria			

## Data Availability

The data presented in this study are available on request from the corresponding author.

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
