# Peer review of "Adaptation of Brucella melitensis Antimicrobial Susceptibility Testing to the ISO 20776 Standard and Validation of the Method"

_microorganisms, 2022, doi:10.3390/microorganisms10071470_

Round 1

Reviewer 1 Report

In this manuscript the authors adapted CLSI recommendations for Brucella melitensis antimicrobial susceptibility testing to the ISO 20776 standard regarding the culture medium and the incubation conditions. They devised a Standard Operation Procedure and performed an interlaboratory validation of the method involving eight different laboratories.

I suggest the manuscript would be suitable for publication in Microorganisms following some minor revisions to the text, as follows:

Line 63.  Change “rates in 2008” to “rate in 2008”

Line 131.  Change “B. melitensis 57” to “A total of 57 B. melitensis”

Line 134.  Remove “respectively”

Line 138.  Insert “the” before “National”

Line 152.  Insert “the” before “pH”

Line 179.  Insert “the” before “BMD”

Line 195.  Insert “the” before “EUCAST”

Line 209.  Insert “the” before “BMD”

Line 223.  Insert “the” before “reference”

Line 362.  Insert “the” before “reference”

Line 381.  Change “rage” to “range”

Line 419.  Change “The agent” to “B. melitensis”

Line 426.  Insert a full-stop after “[32,33]”

Line 428.  Italicise “influenza”

Lines 439-441.  Rewrite “Furthermore, regarding CLSI B. melitensis WT MICs for trimethoprim-sulfamethoxazole cluster around the CLSI M45 breakpoint of 2 mg/L”

Line 443.  Specify how long?

Line 469.  Insert “the” before “literature”

Line 472.  Change “as” to “than”

Line 488.  Insert “an” before “alternative”

Lines 489-491.  Rewrite “The data at different time points indicated a reduction of the incubation time to less than 48 h made reading of BMD plates very difficult which invalidated results”

Line 509.  Change “graduate” to “gradual”

Line 518.  Insert “the” before “final”

Line 524.  Insert “the” before “correct”

Reviewer 2 Report

This is an interesting study describing the coordinated efforts of a working group aiming to improve and validate the methodology used for B. melitensis antimicrobial testing.

The manuscript is well-written and organized. The methodology along with the results of the research are presented explicitly. The literature review is thorough.

The concept and the findings of this study should hold the attention of clinicians and microbiologists. 

Reviewer 3 Report

This is a thorough study that has evaluated, various culture media typically used for antimicrobial susceptibility testing and culture conditions in order to develop a proper method for the antimicrobial susceptibility testing of the highly pathogenic bacterium B. melitensis. It is of the benefit of the international scientific community that new improved protocols are continuously evaluated and evolved and from this point the findings of this study are important.

Some suggestions for improvement:

Line 53 and Line 116: “Muller-Hinton” please change to Mueller-Hinton

Figure 2: It would be helpful for the reader if you could include a small legend above each set of graphs, eg for a)  B. melitensis reference strain ATCC 23456  and (b) B. melitensis clinical isolate Bm150048

Figures 4-6: The same applies in these graphs, small legends in the graphs would improve readability
